# Role of Human DNA Ligases in Mediating Pharmacological Activities of Flavonoids

**DOI:** 10.3390/ijms26041456

**Published:** 2025-02-10

**Authors:** Daekyu Sun, Vijay Gokhale

**Affiliations:** 1Department of Pharmacology and Toxicology, College of Pharmacy, University of Arizona, Tucson, AZ 85721, USA; 2Bio5 Institute, University of Arizona, Tucson, AZ 85721, USA; gokhale@reglagene.com; 3Reglagene, Inc., Tucson, AZ 85719, USA

**Keywords:** flavonoids, human DNA ligases, human DNA ligase I, anticancer

## Abstract

Dietary flavonoids are a group of polyphenol compounds originating from plants that have drawn much attention in the last few decades. Flavonoid-rich foods and dietary supplements are used worldwide due to their health benefits, including antioxidative, anti-inflammatory, immunity-enhancing, anticarcinogenic, estrogenic, and favorable cardiovascular effects. The main objective of our study was to explore the molecular targets of flavonoids to gain insight into the mechanism of action behind their biological effects. In this study, a novel class of resorcinol-based flavonoid compounds was identified as a potent inhibitor of human DNA ligase activity. Human DNA ligases are crucial in the maintenance of genetic integrity and cell fate determination. Thus, our results strongly suggest that this activity against human DNA ligases is responsible, at least in part, for the cellular effects of flavonoid compounds. We anticipate that the results from our studies will improve our understanding of how interactions with human DNA ligases cascade into the recognized health benefits of flavonoids, particularly their wide variety of anticancer effects.

## 1. Introduction

Dietary polyphenols, which are ubiquitously present in fruits, vegetables, seeds, and drinks, have received much recent attention for their health benefits [1,2,3]. More than 8000 different types of polyphenols have been identified so far and divided into ten different classes based on their chemical structure [1,2,3]. The most abundantly occurring polyphenols in plants include flavonoids, phenolic acids, stilbenes, and lignans. The flavonoids are the most studied group of plant polyphenols and account for 60% of ~4000 dietary polyphenols [1,2,3]. As shown in Figure 1, flavonoids are structurally characterized by two benzene rings joined by a pyran ring, or a similar structure of three rings, which are referred to as A, B, and C, respectively [3].

The A ring of most flavonoids has a resorcinol moiety containing 5,7-OH groups. The B ring for different flavonoids varies by the number of hydroxyl groups. For example, luteolin and quercetin have 3′,4′-OH groups in their B ring, while compounds such as epigallocatechin and myricetin contain three hydroxyl groups. The heterocyclic C ring differs across flavonoids in the conjugation of rings and the position of hydroxyl, methoxy, and glycosidic groups. These variations in the C ring define six major groups of flavonoids: flavanols, flavones, flavanones, resorcinol, isoflavonoids, and anthocyanidins (see Figure 1).

The health benefits of dietary flavonoids make them promising candidates for the treatment and prevention of various human diseases, such as neoplastic, inflammatory, immune, cardiovascular, and metabolic diseases [1,2,3,4]. However, it is difficult to find a molecular basis for this positive correlation between flavonoid intake and reduced disease risk [3,4]. Several recent studies have demonstrated that, depending on their structure, flavonoids could be potent inhibitors of several enzymes, including GTPase, protein kinase C, tyrosine kinase, phosphatidylinositol 3-kinase, and topoisomerase II; they also inhibit the activity of membrane transport proteins, mRNA metabolism/alternative splicing proteins, and heterogeneous nuclear ribonucleoprotein A2 [1,2,3,4,5]. However, cellular targets and mechanisms of action for flavonoids remain largely undetermined, and further efforts are necessary to understand how dietary flavonoids exert their beneficial effects.

Mammalian DNA ligases play essential roles in DNA replication, recombination, and repair by catalyzing the formation of phosphodiester bonds between adjacent 5′-phosphoryl and 3′-hydroxyl termini at single breaks in duplex DNA molecules [6,7]. DNA ligase I is the enzyme ultimately responsible for sealing single-strand nicks naturally generated by duplex DNA during DNA transactions such as replication and repair [6,7]. While DNA ligase I represents the majority of DNA ligase activity in proliferating cells, other DNA ligases account for most ligase activity in resting cells [8]. DNA ligase III is considered to have a role in meiotic recombination in the testes and some repair pathways in mammalian cells [9]. DNA ligase IV is responsible for the ligation step in V[D]J recombination in lymphoid cells and the repair of DNA double-strand breaks in mammalian cells [10,11]. DNA ligase activity is required to repair DNA strand breaks generated by normal DNA transactions and to prevent the continued formation of single-strand breaks [SSBs] and double-strand breaks [DSBs]; for example, Okazaki fragments are joined by DNA ligase I during DNA repair and lagging-strand DNA synthesis [12].

In our previous study, a new rapid assay method for DNA ligases was developed to allow direct quantification of enzyme activity without using traditional polyacrylamide gel electrophoretic techniques [13]. This new method allowed us to identify resorcinol derivatives from a commercial chemical library as human DNA ligase I inhibitors [13]. Since the resorcinol moiety in the A rings of dietary flavonoids shares structural similarity with known human DNA ligase inhibitors, the selected flavonoids were tested for potential activity against DNA ligases in this study.

## 2. Results

In this study, a standard nick-sealing assay was used to identify potential human DNA ligase inhibitors from representative flavonoids [13]. As shown in Figure 2, this assay has been very effective in identifying potential human DNA ligase inhibitors such as resorcinol derivative LI-01 (see also a plot in Appendix A) [13]. Selected flavonoids with the resorcinol moiety in their A rings were tested as potential DNA ligase inhibitors. Epigallocatechin gallate (EGCG) is also a flavonoid found in plants, herbs, fruits, and green tea [1,2,3,4]. Rutin is a flavonoid glycoside combining the flavonol quercetin and the disaccharide rutinose found in a wide variety of plants, including citrus [1,2,3,4]. Quercetin is a plant flavonoid found in many fruits, vegetables, and other foods [1,2,3,4]. Numerous studies using both in vitro and in vivo models of various tumors have demonstrated that both flavonoids can exert anti-tumor functions through various mechanisms [1,2,3,4,5].

As shown in Figure 3, representative flavonoids, including epigallocatechin, quercetin, and rutin, turned out to be potent inhibitors of human DNA ligase I based on our biochemical assay, with IC_50_ values of 0.5, 0.5, and 3.0 μg/mL, respectively (see also a plot in Appendix A).

Cyanidin, a flavonoid with free hydroxyl groups on its resorcinol and catechol rings, is a type of anthocyanidin found in many red fruits and vegetables [1,2,3,4]. Rosinidin is an anthocyanidin cation consisting of benzopyrylium with hydroxy substituents at positions 3 and 5, a methoxy group at position 7, and a 4-hydroxy-3-methoxyphenyl group at position 2 [14]. As shown in Figure 4, cyanidin also showed potent inhibitory activity against human DNA ligase I at an IC_50_ value of about 1.0 µg/mL (see also plots in Appendix A). However, rosinidin showed little activity against human DNA ligase I, suggesting that unsubstituted hydroxyl groups on the resorcinol ring are required to mediate inhibitory activity.

We further investigated the possibility of direct interactions between selected flavonoids and DNA ligase I through molecular modeling studies. The X-ray crystal structure of DNA ligase I with partially unfolded, nicked DNA was available from the Protein Databank [www.rcsb.org: pdb code: 1X9N] [15]. As shown in Figure 5, a docking analysis of quercetin with an AMP-binding site on DNA ligase I was performed using Glide within the Schrodinger suite of programs [16]. Quercetin binds the adenylation domain such that its catechol ring forms two hydrogen bonds with Glu566 and Tyr567 [backbone amide] residues, while its resorcinol ring extends into and interacts near the pocket created by Glu621. The catechol ring is also stacked between Phe660 and Met723 within the pocket. An adenylation domain is also present in other DNA ligases, strongly implying that flavonoids would also interact with DNA ligases III and IV.

To validate the results from the molecular modeling studies, we also tested the effects of rutin on human DNA ligase III and IV activity. As shown in Figure 6, rutin inhibited the activity of both DNA ligases with IC_50_ values of 0.5 and 1.0 μg/mL, respectively (see also a plot in Appendix A).

Some flavonoids are known to be cytotoxic to different types of human cancer cell lines [17,18,19]. In this study, Caco-2 cells were exposed to rutin at concentrations lower than IC_50_ (25 μg/mL) for 48 h, and images were taken using phase contrast microscopy. Fewer cells were present in rutin-treated cultures after 48 h incubation, confirming its inhibitory effect on cell growth (Figure 7A). Phase contrast views reveal that rutin-treated cells are relatively thin and shrunken, consistent with the reduced cellular densities commonly observed in cultures exposed to cytotoxic agents (Figure 7A). Our data also revealed that rutin increases levels of γH2AX protein in a dose-dependent manner (Figure 7B). The γH2AX is a sensitive and selective marker for DSBs, as phosphorylation of H2AX (γH2AX) is increased at such breakage sites [20]. This suggests the DNA-damaging potential of flavonoids through the inhibition of DNA ligases.

## 3. Discussion

Flavonoids are natural components in our diet, and flavonoid preparations are marketed as herbal medicines or dietary supplements for a variety of alleged therapeutic effects [1,2,3]. However, their potential for health benefits and toxicity is an understudied field of research [1,2,3]. There is an increasing interest in understanding the mechanisms of action for the cellular effects of flavonoids, especially since many of their cellular targets remain undetermined [1,2,3]. Identifying such cellular targets, such as human DNA ligases, is a challenging but essential step to understanding how biological interactions with flavonoids cascade into their recognized health benefits.

As described in this study, a novel class of resorcinol-based flavonoid compounds has been identified as potent DNA ligase I inhibitors. DNA ligase I is the enzyme ultimately responsible for sealing single-strand nicks naturally generated by duplex DNA during DNA transactions such as replication and repair [6,7]. Therefore, interference with DNA ligase I activity could directly or indirectly result in the generation of single stranded breaks (SSBs) [12]. SSBs that are improperly repaired may result in double stranded breaks (DSBs), increasing genomic damage and leading to the activation of DNA repair and damage signaling pathways [12]. Considering the essential role of DNA ligases in DNA replication, recombination, and repair [1,2], we propose that the biological effects of flavonoids can be attributed, at least in part, to their activity against DNA ligases.

Our previous studies suggested that DNA ligase I is upregulated by DNA-damaging chemotherapeutic agents [e.g., gemcitabine, irinotecan, and cisplatin] through the activation of DNA checkpoint pathways involving ATR-Chk1 [21,22,23]. Other studies have clearly identified DNA ligase I inhibitors as therapeutic anticancer agents and potential modulators for the therapeutic efficacy of standard chemotherapeutic agents [24,25,26,27]. Thus, the significance of our study is in offering a mechanism of action for the beneficial effects of flavonoids on cancer chemotherapy, specifically through inhibition of DNA repair via interference with DNA ligase I activity.

DNA repair pathways can enable cancer cells to enhance their DNA repair capacity and survive DNA damage induced by chemotherapy, allowing them to survive and acquire resistance to treatment [28]. Conversely, deficient DNA repair capacity in cancer cells may allow DNA-damaging chemotherapeutics to cause unrepairable and more cytotoxic DNA lesions [28]. DNA repair inhibitors such as DNA ligase inhibitors can enhance the effects of DNA-damaging cancer drugs, particularly against cancer cells with defective DNA repair pathways [28]. For example, PARP inhibitors, such as Olaparib (Lynparza), have been shown to enhance the effects of other DNA-damaging drugs, such as gemcitabine [29].

Similarly, high doses of quercetin were found to synergistically potentiate the anticancer activity of DNA-damaging drugs [30]. Epigallocatechin gallate (EGCG) was also found to induce DNA damage in cancer cells, potentially enhancing the effects of certain chemotherapeutic drugs [31]. EGCG may also cause DNA damage in normal cells at high concentrations [31]. A previous study also suggested that cyanidin can sensitize a gemcitabine-resistant cholangiocarcinoma cell line to gemcitabine treatment [32].

Thus, we hypothesized that flavonoids with DNA ligase inhibitory activity would enhance the cytotoxicity of DNA-damaging anticancer drugs in human cancer cells. The combination of DNA-damaging anticancer agents with flavonoids with DNA inhibitory activity could overcome drug resistance, potentially increasing the therapeutic efficacy of cancer chemotherapy by inhibiting DNA repair-mediated removal of toxic DNA lesions.

Further research is critical to verify whether inhibition of DNA ligase activity by dietary flavonoids correlates with their cellular effects, as well as understand whether inhibitory activity is mediated by direct inhibition of the DNA ligases themselves. We anticipate that the results from our proposed studies will improve our understanding of how interactions between flavonoids and cellular DNA ligases cascade into recognized health effects, particularly the broad antitumor effects of dietary flavonoids.

## 4. Materials and Methods

### 4.1. Materials

The ligase inhibitor 2-(3,5-dihydroxypennyl)-2-methyldecalin (LI-01) was purchased from Microsource Discovery Systems, (Gaylordsville, CT, USA). All flavonoids used in this study were purchased from Sigma-Aldrich, Inc. (St. Louis, MO, USA). All oligonucleotides were obtained from Sigma Genosys Biotechnologies (Woodlands, TX, USA). Beta actin antibody (sc-47778) and p-histone H2AX antibody (sc-517348) were purchased from Santa Cruz Biotechnology (Dallas, TX, USA). The Caco-2 cell line was obtained from the American Type Culture Collection (Rockville, MD, USA). 

### 4.2. Preparation of DNA Substrate for DNA Ligases

A 39 base pair nicked duplex was constructed and used as a substrate for the ligase reaction. In brief, the first 19 base pair oligonucleotide (5′-CGTAAAACGACGGCCAGTG-3′) and the second 20 base pair oligonucleotide (5′-AATTCGAGCTCGGTACCCGG-3′) labeled with radioactive ^32^P were annealed to the complementary 39 base pair oligonucleotide (5′-CCGGGTACCGAGCTCGAATTCACTGGCCGTCGTTTTACG-3′) to construct a 39 base pair nicked duplex as previously described [13]. The resulting nicked duplex DNA was purified via 8% nondenaturing polyacrylamide gel electrophoresis [13].

### 4.3. DNA Ligase Assay Using Electrophoresis

DNA ligase I activity was routinely assayed according to a modified procedure described previously [13] by determining the ligation of two oligonucleotides, which are hybridized to a complementary 39-mer oligonucleotide. Ligation of 5′-[^32^P] end-labeled 20-mer oligonucleotide to the unlabeled 19-mer oligonucleotide was analyzed via polyacrylamide gel electrophoresis in the presence of 8 M urea. The assays were performed at least twice, and representative data are shown in this paper.

### 4.4. Molecular Modeling

The X-ray crystal structure of DNA ligase I with partially unfolded, nicked DNA was obtained from the Protein Databank [www.rcsb.org: pdb code: 1X9N] [15]. The docking analysis of quercetin with an AMP-binding site on DNA ligase I was performed using Glide within the Schrodinger suite of programs [16].

### 4.5. Assessment of Cell Shape and Morphology

The human intestinal Caco-2 cells were grown at 37 °C and 5% CO_2_ in a humidified atmosphere, and treated and untreated control cells were examined using an inverted phase-contrast microscope (Fisher Scientific Inverted Phase Contrast Microscope EQ-006, Pittsburgh, PA, USA) to assess cell shape and morphology.

## Figures and Tables

**Figure 1 ijms-26-01456-f001:**
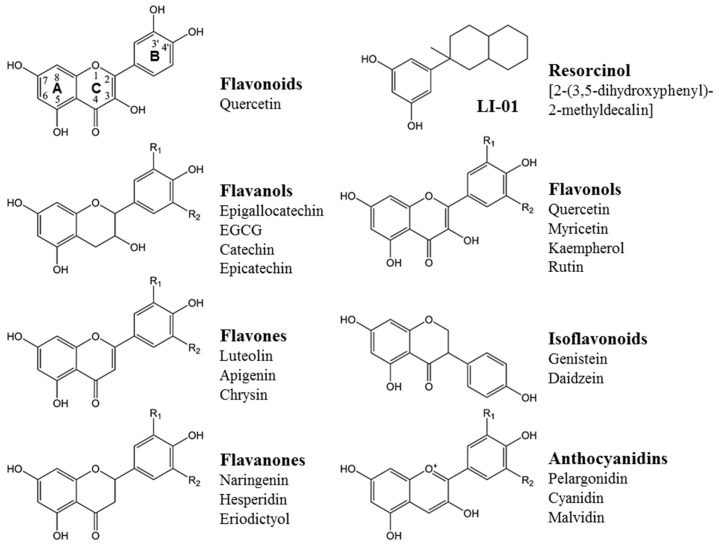
Basic chemical structures of commonly occurring flavonoids present in major dietary sources.

**Figure 2 ijms-26-01456-f002:**
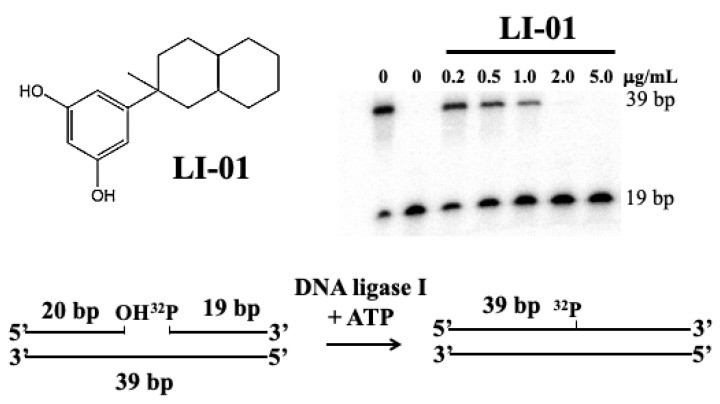
The structure of resorcinol derivative LI-01 and its effect on human DNA ligase I activity measured via a nick-sealing assay using a nicked duplex oligomer DNA. DNA ligase assays were performed in reaction mixtures in the presence of ATP (10 μM), DNA substrate (50 nM) and human DNA ligase I 90.5 nM) as previously described [13].

**Figure 3 ijms-26-01456-f003:**
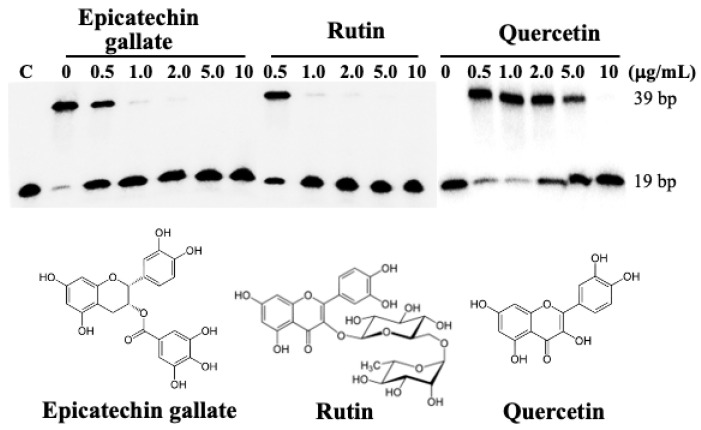
The effects of selected flavonoids on human DNA ligase I activity. DNA ligase assays were performed in reaction mixtures in the presence of ATP (10 μM), DNA substrate (50 nM), and human DNA ligase I (0.5 nM) as previously described [13].

**Figure 4 ijms-26-01456-f004:**
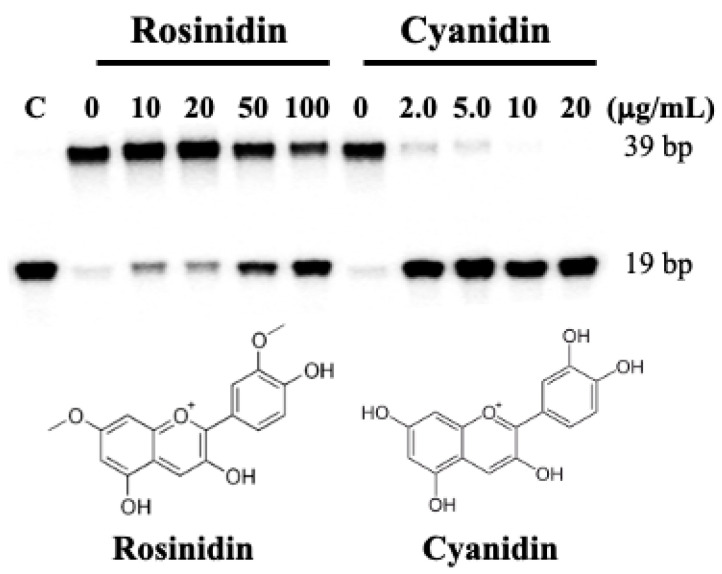
The effects of cyaniding and rosinidin on human DNA ligase I activity. DNA ligase assays were performed in reaction mixtures in the presence of ATP (10 μM), DNA substrate (50 nM), and human DNA ligase I (0.5 nM) as previously described [13].

**Figure 5 ijms-26-01456-f005:**
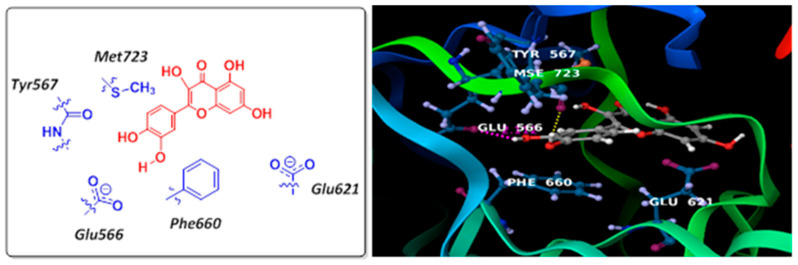
Docking model of quercetin with human DNA ligase I adenylation domain (right panel) and schematic representation (left panel).

**Figure 6 ijms-26-01456-f006:**
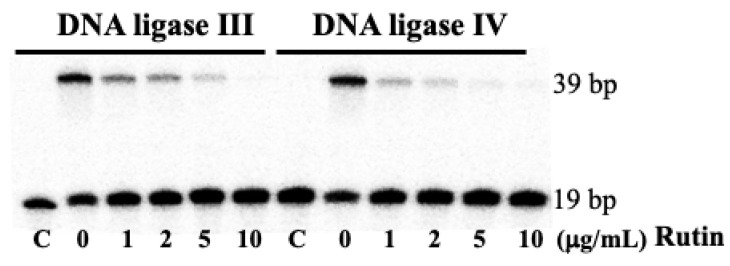
The effects of rutin on human DNA ligases III and IV activities. DNA ligase assays were performed in reaction mixtures in the presence of ATP (10 μM), DNA substrate (50 nM), and human DNA ligase I (0.5 nM) as previously described [13].

**Figure 7 ijms-26-01456-f007:**
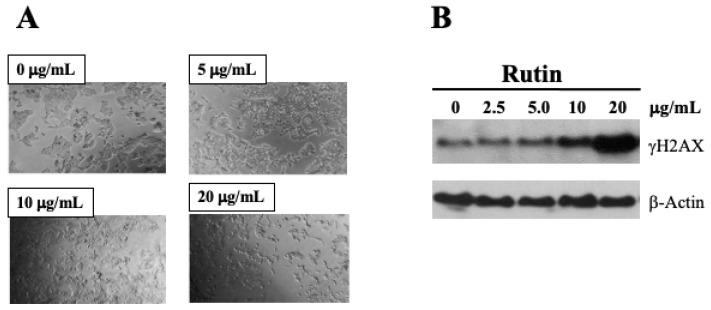
Distinct morphological changes of Caco-2 cells in response to rutin treatment for 48 h. Phase contrast micrographs of Caco-2 cells treated with various concentrations of rutin for 48 h. Photographs taken at 10× magnification: (**A**) increased expressions of γH2AX protein in Caco-2 cells incubated with rutin for 48 h, compared with corresponding solvent control group, as measured via Western blot (**B**).

## Data Availability

Data is contained within the article and Appendix A.

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
