# Peer review of "Role of Human DNA Ligases in Mediating Pharmacological Activities of Flavonoids"

_ijms, 2025, doi:10.3390/ijms26041456_

Round 1

Reviewer 1 Report

Comments and Suggestions for Authors

I reviewed the manuscript entitled Role of human DNA ligases in mediating pharmacological activities of flavonoids.

 I agree to accept this manuscript after major revision. 

1)  Keywords: Flavonoids; Human DNA ligases; Human DNA ligase I; Anticancer Except for the capitalization of the first letter of the first keyword, the first letter of the first word of other keywords does not need to be capitalized.

2) Suggest the author to mark the ABC rings of Quercetin in Figure 1, so that readers can see these three rings more intuitively.

3) Figure 2. as previously described [13]. . Delete a period.

4) Figure 3. Quercetin, the concentration and bands are not aligned, please modify them.

5) In scientific papers, avoid using words such as' we 'and' our ', and maintain rigor and objectivity.

6) In this study, Caco-2 cells were exposed to rutin at concentrations lower than IC50 (25 g/mL) for 48h, 48h should change to 48 h. There should be spaces between numbers and units, except for % and oC. Please revise the entire text according to this standard.

7) γH2AX, γ should be italicized, all Greek letters need to be italicized, check and modify the entire text.

8) Figure 7. 48 hours should change to 48 h. Figure 7B, rH2AX should change to γH2AX, b should be italicized.

9) Supplementary Materials: Npne should change to None.

10) I have read all the references and found some issues. All journal names of references need to be italicized, years need to be bolded, and volumes need to be italicized. Ref 6, the author's format is incorrect. Ref 14, Catharanthus roseus should be italicized. The author should check and revise the references according to the requirements of the journal.

11) More than 8,000 different types of polyphenols have been identified so far and are divided into ten different classes based on their chemical structure [1-3]. Refs 1-3, the latest of which has been published for more than ten years, are far from meeting the novelty requirements. In my impression, more than ten thousand polyphenols have been isolated. Please conduct a new search and update the references.

12) The study explored the molecular targets of flavonoids, a group of polyphenol compounds from plants known for health benefits. A novel class of resorcinol-based flavonoid compounds was identified as potent inhibitors of human DNA ligase activity, crucial for genetic integrity and cell fate. Our results suggest this activity contributes to flavonoids' cellular effects, enhancing understanding of their health benefits, especially anticancer effects.

13) The biggest problem with this article is that many of the references cited are too outdated, which may result in insufficient novelty or incorrect conclusions. Therefore, the author must make necessary updates to them to ensure that my concerns do not arise.

14) The other published materials on this topic primarily concentrate on the pharmacological effects of flavonoids, encompassing antioxidant, anti-inflammatory, immune-enhancing, anticarcinogenic, estrogenic, and cardioprotective properties. The primary objective of this study was to delve into the molecular targets of flavonoids in order to gain insights into the mechanisms underlying their biological effects. This research identified a novel class of resorcinol-based flavonoid compounds as potent inhibitors of human DNA ligase activity.

Author Response

A point-by-point response to the reviewer # 1's comments and concerns

 We greatly appreciate Reviewer #1 who has taken their valuable time to contribute to the reviewing process of our article.

1)  Keywords: FlavonoidsHuman DNA ligasesHuman DNA ligase IAnticancer Except for the capitalization of the first letter of the first keyword, the first letter of the first word of other keywords does not need to be capitalized.

The authors thank the reviewer’s comment. We have altered it accordingly.

2) Suggest the author to mark the ABC rings of Quercetin in Figure 1, so that readers can see these three rings more intuitively.

As the reviewer suggested, we have altered it accordingly.

3) Figure 2. as previously described [13]. . Delete a period.

The authors thank the reviewer’s comment. As the reviewer suggested, we have altered it accordingly.

4) Figure 3. Quercetin, the concentration and bands are not aligned, please modify them.

As the reviewer suggested, we have altered it accordingly.

5) In scientific papers, avoid using words such as' we 'and' our ', and maintain rigor and objectivity.

As the reviewer suggested, we have altered it accordingly.

6) In this study, Caco-2 cells were exposed to rutin at concentrations lower than IC50 (25 g/mL) for 48h, 48h should change to 48 h. There should be spaces between numbers and units, except for % and oC. Please revise the entire text according to this standard.

As the reviewer suggested, we have altered it accordingly.

7) γH2AX, γ should be italicized, all Greek letters need to be italicized, check and modify the entire text.

As the reviewer suggested, we have altered it accordingly.

8) Figure 7. 48 hours should change to 48 h. Figure 7B, rH2AX should change to γH2AX, b should be italicized.

As the reviewer suggested, we have altered it accordingly.

9) Supplementary Materials: Npne should change to None.

As the reviewer suggested, we have altered it accordingly.

10) I have read all the references and found some issues. All journal names of references need to be italicized, years need to be bolded, and volumes need to be italicized. Ref 6, the author's format is incorrect. Ref 14, Catharanthus roseus should be italicized. The author should check and revise the references according to the requirements of the journal.

As the reviewer suggested, we have altered it accordingly.

11) More than 8,000 different types of polyphenols have been identified so far and are divided into ten different classes based on their chemical structure [1-3]. Refs 1-3, the latest of which has been published for more than ten years, are far from meeting the novelty requirements. In my impression, more than ten thousand polyphenols have been isolated. Please conduct a new search and update the references.

As the reviewer suggested, we added updated references (please see the highlighted parts of the reference. 

12) The study explored the molecular targets of flavonoids, a group of polyphenol compounds from plants known for health benefits. A novel class of resorcinol-based flavonoid compounds was identified as potent inhibitors of human DNA ligase activity, crucial for genetic integrity and cell fate. Our results suggest this activity contributes to flavonoids' cellular effects, enhancing understanding of their health benefits, especially anticancer effects.

The authors thank the reviewer’s comment.

13) The biggest problem with this article is that many of the references cited are too outdated, which may result in insufficient novelty or incorrect conclusions. Therefore, the author must make necessary updates to them to ensure that my concerns do not arise.

As the reviewer suggested, we added updated references (please see the highlighted parts of the reference. 

14) The other published materials on this topic primarily concentrate on the pharmacological effects of flavonoids, encompassing antioxidant, anti-inflammatory, immune-enhancing, anticarcinogenic, estrogenic, and cardioprotective properties. The primary objective of this study was to delve into the molecular targets of flavonoids in order to gain insights into the mechanisms underlying their biological effects. This research identified a novel class of resorcinol-based flavonoid compounds as potent inhibitors of human DNA ligase activity.

The citation was altered accordingly.

The authors thank the reviewer’s comment.

Reviewer 2 Report

Comments and Suggestions for Authors

The manuscript described an interesting work to address potential pharmacological properties of flavonoids on DNA ligase activity, but need an improuvement on data presentation and discussion of the results. 

In detail:

- Result section should be more clear, maybe if starting with the explanation of the selected flalvonoids. Now there are sentences that are repeated from the last part of the Introduction to the first lines of the Results. 

- Figures 2, 3, 4 and 6 should be improuved. Check also the allignment of each Figure and insert A,B...for each panel, by explaining them better in the legends. 

- I don't see any statistical analysis. Why? How many replicates did you perform for each experiments?

- Figure 7: the quality of the images in the panel A should be improuved; add the scale bar and the information about the magnification used.

- Enlarge the discussion section by pointing out the relevance of the used molecules and the reported strategy in comparison with other prevoius studies.

Author Response

A point-by-point response to the reviewer # 2's comments and concerns

 We greatly appreciate Reviewer #2 who has taken their valuable time to contribute to the reviewing process of our article.

As the reviewer suggested, we have altered our manuscript accordingly (please see the highlighted parts and revised figures).

The manuscript described an interesting work to address potential pharmacological properties of flavonoids on DNA ligase activity, but need an improvement on data presentation and discussion of the results. 

In detail:

- Result section should be more clear, maybe if starting with the explanation of the selected flalvonoids. Now there are sentences that are repeated from the last part of the Introduction to the first lines of the Results. 

As the reviewer suggested, we have altered them accordingly (please see the highlighted parts).

- Figures 2, 3, 4 and 6 should be improuved. Check also the allignment of each Figure and insert A,B...for each panel, by explaining them better in the legends. 

As the reviewer suggested, we have altered them accordingly (please see the revised figures).

- I don't see any statistical analysis. Why? How many replicates did you perform for each experiments?

The assays were performed at least twice, and representative data are shown in this paper.

- Figure 7: the quality of the images in the panel A should be improved; add the scale bar and the information about the magnification used.

Unfortunately, we missed the scale bar, but information about the magnification used is included.

- Enlarge the discussion section by pointing out the relevance of the used molecules and the reported strategy in comparison with other previous studies.

As the reviewer suggested, we have altered discussion part of our manuscript accordingly (please see the highlighted parts in Discussion).

Round 2

Reviewer 1 Report

Comments and Suggestions for Authors

The author has made the necessary modifications and explanations as per my request. But there are still three issues that need to be addressed.

1) 4.5. Assessment of cell shape and morphology, 37 °C should change to 37°C.

2) The format of the literature has not been modified according to my previous suggestion, please make the necessary changes.

3) Ref 31, Camellia sinensis requires italics.

Reviewer 2 Report

Comments and Suggestions for Authors

The manuscript has been improuved. However, there are still some points to be addressed:

- You say that: The assays were performed at least twice, and representative data are shown in this paper. 

If you have more experimental data could you provide a graph with the representation of statistical numbers? I think this should be possible by representation of the enzymatic activity (e.g. % of ligation?), as you describe in the original paper of this method. I think it is necessary to improuve the scientific relevance of the manuscript. You can insert these data in the resuls section or at least as supplementary material.

- If you have neither the error bar of the images in the Figure 7A and you can not improuve the quality of the pictures, I think that it should be remouved. 

Round 3

Reviewer 2 Report

Comments and Suggestions for Authors

The manuscript has been improuved. I appreciate the insertion of the Supplementary data (even if I should prefere the indications about number of replicates (n) other than "at least two replicates" and the use of a statistical analysis.